# Association Between Hyperlipidaemia and Selected Cholestatic Markers in 74 Dogs with Suspect Acute Pancreatitis

**DOI:** 10.3390/ani14223281

**Published:** 2024-11-14

**Authors:** Adrien J. Da Silva, Aimee Hope, Carmel T. Mooney

**Affiliations:** 1University College Dublin Veterinary Hospital, University College Dublin, D04W6F6 Dublin, Ireland; carmel.mooney@ucd.ie; 2VetsNow Referrals, Glasgow G3 7DA, UK; aimee.hope@btinternet.com

**Keywords:** hypercholesterolaemia, hypertriglyceridaemia, cholestasis, pancreatic disease

## Abstract

The cause of hyperlipidaemia in dogs with acute pancreatitis remains unknown. Cholestasis potentially results in increased circulating cholesterol concentrations in dogs with acute pancreatitis, but no study has specifically investigated such a relationship, and the possibility of other causes has been suggested. The aim of this study was to investigate the association between hyperlipidaemia and other circulating markers of cholestasis (alkaline phosphatase (ALP) and gamma-glutamyl-transferase (GGT) activities) in dogs with acute pancreatitis. Cases of acute pancreatitis were reviewed and divided into two groups: dogs with increased cholesterol concentrations and dogs with normal cholesterol concentrations. Dogs with increased cholesterol concentrations had significantly higher ALP activities compared to dogs with normal cholesterol concentrations. There was a significant positive correlation between the cholesterol concentration and ALP and gamma-glutamyl transferase (GGT) activities. Such results support the hypothesis that cholestasis plays a role in the development of hypercholesterolaemia in dogs with acute pancreatitis.

## 1. Introduction

Hyperlipidaemia is a general term that refers to increased circulating concentrations of lipids, including both triglyceride- and cholesterol-rich lipoproteins. Hypertrigyceridaemia and hypercholesterolaemia are reported in dogs with acute pancreatitis albeit infrequently (18 and 24%, respectively) if secondary causes of hyperlipidaemia and prior drug therapies are excluded [1].

Hypertriglyceridaemia is a well-known risk factor for pancreatitis in humans [2], but the relationship between hyperlipidaemia and acute pancreatitis in dogs is unclear. Experimental studies identified that hyperlipidaemia was an inconsistent feature in experimentally induced acute pancreatitis in dogs [3,4,5]. A previous clinical study suggested that hypertriglyceridaemia could increase the risk of acute pancreatitis in miniature schnauzers: dogs with a history of pancreatitis were five times more likely to have persistent hypertriglyceridaemia after resolution of the pancreatitis [6].

It is believed that cholestasis could play a role in the development of hyperlipidaemia in dogs with acute pancreatitis. A high prevalence (93%) of hypercholesterolaemia (up to 30 mmol/L) has been reported in dogs presenting with extrahepatic biliary obstruction and pancreatitis [7]. Cholestasis is also a possible contributor to hypertriglyceridaemia in dogs but expected increases are mild [8,9]. Nevertheless, other potential mechanisms for hyperlipidaemia in acute pancreatitis are possible, and such a diagnosis is now often viewed as a separate differential diagnosis for hyperlipidaemia, distinct from other cholestatic diseases [10].

The aim of this study was to investigate the association between cholesterol concentration and other markers of cholestasis in dogs with suspected acute pancreatitis.

## 2. Materials and Methods

Case records of dogs seen between January 2014 and June 2022 that had a qualitative point-of-care canine pancreatic lipase (cPL) (SNAP cPL™, IDEXX Laboratories, Westbrook, ME, USA) charged were retrospectively reviewed. In these cases, results are classified as normal (corresponding to a quantitative cPL concentration of <200 μg/L), abnormal (quantitative cPL ≥ 200 μg/L) or equivocal (not possible to confidently place into either of these categories). With either of the latter two results, quantitative cPL measurement is carried out (Spec cPL™, IDEXX Laboratories, Wetherby, UK). If the point-of-care cPL was not recorded, the case was only included if a quantitative cPL result was available.

A diagnosis of suspect acute pancreatitis was made if there were either ultrasonographic changes suggesting acute pancreatitis (i.e., pancreatic enlargement, hypoechogenicity of the parenchyma, ill-defined margins, hypoechoic stripes and anechoic lesions, hyperechogenicity of the peripancreatic fat) or, the quantitative cPL was ≥400 µg/L and both the clinician-stated diagnosis was pancreatitis and the treatment qualified the disease as acute (intravenous fluid therapy, analgesia), as previously described (Hope et al. 2021). Dogs with incomplete biochemical data were excluded. Dogs that were known to have pre-existing disorders or drug therapies associated with hyperlipidaemia or hypocholesterolaemia were also excluded because of the possibility of exacerbating or masking the changes associated with pancreatitis. Specifically testing all dogs for non-pancreatitis causes of lipid abnormalities was not required but was at the discretion of the attending clinician. Dogs were divided into two groups: dogs with hypercholesterolaemia (HC group) and dogs without hypercholesterolaemia (NC group). Signalment, and biochemical and haematological data on admission were recorded.

Continuous data were assessed for normality using the Shapiro–Wilk method. Data were reported as median (interquartile range, IQR) or mean ± standard deviation (sd), as appropriate. For variables not normally distributed, a Mann–Whitney U test was used for comparison. Spearman’s rank correlation coefficient was used to assess correlation between two variables. The strength of correlation was interpreted as r_s_ = 0–0.19, very weak; 0.2–0.39, weak; 0.4–0.59, moderate; 0.6–0.79, strong 0.8–1, very strong. Chi-squared analysis was used for categorical data. Statistical significance was set at *p* < 0.05. All statistical analyses were performed using GraphPad Prism (version 10.3.1 GraphPad Software LLC, Boston, MA, USA).

## 3. Results

In total, 889 dogs were identified, but only 213 cases fulfilled the criteria of suspect acute pancreatitis. A further 139 (65.3%) cases were excluded because of pre-existing disorders or drug therapies associated with hyperlipidaemia (*n* = 115), hypocholesterolaemia (*n* = 22) and incomplete biochemical data (*n* = 2). The remaining 74 dogs included 17 crossbreeds and 57 purebreds of 27 different breeds: five each of Bichon Frise and Yorkshire Terrier, four each of Cocker Spaniel, Jack Russel Terrier, Miniature Schnauzer, three each of Boxer, Golden Retriever, Rottweiler, two each of Bearded Collie, Cavalier King Charles Spaniel, French Bulldog, Labrador Retriever, Lurcher, Shih Tzu, Springer Spaniel and one each of Border Collie, Chihuahua, Dalmatian, German Shepherd, Glen of Imaal Terrier, Irish Wolfhound, Lagotto Romagnolo, Maltese, Pomeranian, Staffordshire Bull Terrier, Tibetan Terrier, West Highland White Terrier. Age, sex distribution, body weight and body condition score are presented in Table 1. There was no significant difference between groups in terms of age, sex distribution, and body weight (*p* = 0.847; *p* = 0.915 and *p* = 0.728, respectively). Body condition score was significantly (*p* < 0.001) higher in the HC compared to the NC group.

The biochemical results are presented in Table 2. There were 33 (44.6%) dogs with hypercholesterolaemia and 41 (56.4%) without. Triglyceride concentrations were increased in 17 (23.0%) dogs. Triglyceride concentrations were significantly (*p* < 0.001) higher in the HC compared with the NC group. However, although more dogs in the HC group had hypertrigyceridaemia compared with the NC group (*n* = 13, 39.4% and *n* = 4, 9.8%, respectively; *p* = 0.003), no value exceeded 5 mmol/L.

Seventy dogs (94.6%) had increased ALP activity. All dogs in the HC group had increased ALP activity. None of the remaining four dogs with an ALP activity within reference interval had hypercholesterolaemia. The proportion of dogs with increased ALP activity was not significantly different between the two groups (*p* = 0.065). The ALP activity was significantly higher (*p* < 0.001) in the HC group compared to the NC group. There was a moderate positive correlation between cholesterol concentration and ALP activity (r_s_ = 0.498, *p* < 0.001) (Figure 1).

Fifteen of 73 (20.5%) dogs in which it was measured had increased GGT activity. Of these, nine (60.0%) had hypercholesterolaemia. Of the remaining 58 dogs, 24 (41.4%) had hypercholesterolaemia, which was not significantly different (*p* = 0.197). There was no significant difference (*p* = 0.303) in GGT activity between the two groups and only a weak positive correlation between cholesterol concentration and gamma-glutamyl transpeptidase (GGT) activity (r_s_ = 0.296, *p* = 0.011).

Nineteen (25.3%) dogs had hyperbilirubinaemia of which 12 (63.2%) had hypercholesterolaemia. Of the remaining 55 dogs, 21 (38.2%) had hypercholesterolaemia, which was not significantly different (*p* = 0.059). There was no significant difference (*p* = 0.999) in total bilirubin concentrations between the two groups. Four dogs had extrahepatic bile duct obstruction, and all had severe hypercholesterolaemia (range: 13.59–21.79 mmol/L), marked hyperbilirubinaemia (range: 81.1–183.8 μmol/L), and severe increases in ALP (range: 7935–21,120 IU/L) and GGT (range: 41–189 IU/L) activities.

Quantitative cPL concentrations were measured in 62 dogs and 1,2-o-dilauryl-rac-glycero-3-glutaric acid-(6′-methylresorufin) ester (DGGR) lipase activity in all dogs. There was no significant difference in quantitative cPL concentration between the NC and HC groups (*p* = 0.122) and DGGR lipase activity (*p* = 0.477). There was no correlation between cholesterol concentration and DGGR lipase activity (*p* = 0.252) and a significant but very weak negative correlation with quantitative cPL concentration (r_s_ = −0.276, *p* = 0.030).

## 4. Discussion

This study aimed to investigate the association between cholesterol and other cholestatic markers in dogs with suspect acute pancreatitis.

Despite excluding cases with other known causes of hyperlipidaemia, approaching half of the dogs with suspect acute pancreatitis in this study had hypercholesterolaemia. This is greater than that of approximately 25% previously reported [1]. The reason for the discrepancy is unclear. The previous study only included 17 dogs, possibly underestimating the prevalence of hypercholesterolaemia in dogs with acute pancreatitis. On the other hand, despite the application of strict exclusion criteria, the current study was retrospective and concurrent causes of hyperlipidaemia could not be definitively ruled out. In addition, in the present study, excluding dogs with concurrent disorders associated with hypocholesterolaemia may also influence results as such exclusion criteria are not always applied in other studies [1]. The body condition score was significantly higher in hypercholesterolaemic dogs. However, although obesity is associated with hypercholesterolaemia, the magnitude of change and its association with other cholestatic abnormalities makes obesity an unlikely explanation for the changes observed in the current study. In addition, the body condition score was not available for all dogs in each of the two groups.

The median ALP activity was significantly higher in dogs with hypercholesterolaemia compared to dogs without hypercholesterolaemia and there was a moderate positive correlation between cholesterol concentrations and ALP activity in dogs with suspect acute pancreatitis. Although there was a weak positive correlation between cholesterol concentration and GGT activity, there was no significant difference in GGT activities between the two groups. Gamma-glutamyl transferase is considered to be a less sensitive marker of hepatobiliary disease in dogs and increases are of a lesser magnitude compared with those in ALP activity [11]. This may explain the differences in the association of hypercholesterolaemia and GGT compared with ALP activities in the current study. Whilst bilirubin concentrations can act as a marker of cholestasis, like GGT, bilirubin is not particularly sensitive and was only increased in approximately one quarter of the dogs in the current study [12]. Thus, the lack of correlation with cholesterol was not unexpected. Nevertheless, hypercholesterolaemia was present in the majority of hyperbilirubinaemic dogs and four of the five highest cholesterol concentrations occurred in the dogs with known extrahepatic biliary obstruction and pancreatitis as previously described [7].

The prevalence of hypertriglyceridaemia of approximately 25% was similar to previous reports [1]. However, as in the previous study, the increases were marginal, with no value exceeding 5 mmol/L reflecting the concentration often used to approximate the therapeutic target in hypertriglyceridaemia [13]. This further supports that the hypertriglyceridaemia associated with acute pancreatitis is mild. Thus, and in accordance with previous recommendations, dogs with moderate to severe hypertriglyceridaemia should be investigated for an alternative concurrent disorder [1]. Acute pancreatitis does not seem to increase triglyceride concentrations to any clinically significant extent.

Both cPL concentrations and DGGR lipase activity are used as markers of pancreatitis with similar diagnostic performance with neither having perfect sensitivity or specificity for acute disease [14]. There was no correlation between cholesterol and DGGR lipase activity and a very weak but negative correlation with cPL concentration. Quantitative cPL concentration and DGGR lipase activity have been shown to be poor indicators of severity, clinical progression or survival in dogs with acute pancreatitis [14,15], possibly explaining any lack of association. 

There are several limitations of this study not least its retrospective nature. Further diagnostic investigations to rule in or out concurrent disorders were at the discretion of the attending clinician. Thus, a concurrent cause of hyperlipidaemia could not be completely excluded in all cases. In particular, dogs were not always specifically investigated for endocrine disorders such as hypothyroidism or hypercortisolism that can be associated with hypertriglyceridaemia, cholestasis, and pancreatitis. However, there appeared to be no clinical indication for such investigations that may in themselves be influenced by the acute nature of such an illness as pancreatitis. No attempt was made to assess the severity of disease because of the complexities associated with its retrospective nature. Follow-up information was not available for most cases, and it is not possible to determine whether the hyperlipidaemia resolved with treatment of the acute pancreatitis. The structure of the study allowed the identification of an association between cholesterol concentrations and specific cholestatic markers, but a cause-and-effect relationship could not be investigated. Further longitudinal studies evaluating cholesterol concentrations over the course of the disease are warranted to further investigate the mechanisms associated with hyperlipidaemia in dogs with acute pancreatitis. 

## 5. Conclusions

In conclusion, the results of this study suggest that there is an association between cholesterol concentration and markers of cholestasis in dogs with acute pancreatitis. Further studies are warranted to investigate the impact of severity and the underlying pathophysiological mechanisms explaining this association. Hypertriglyceridaemia is less likely without hypercholesterolaemia, and when it occurs it is mild. More marked hypertrigyceridaemia should prompt further investigation for a concurrent abnormality.

## Figures and Tables

**Figure 1 animals-14-03281-f001:**
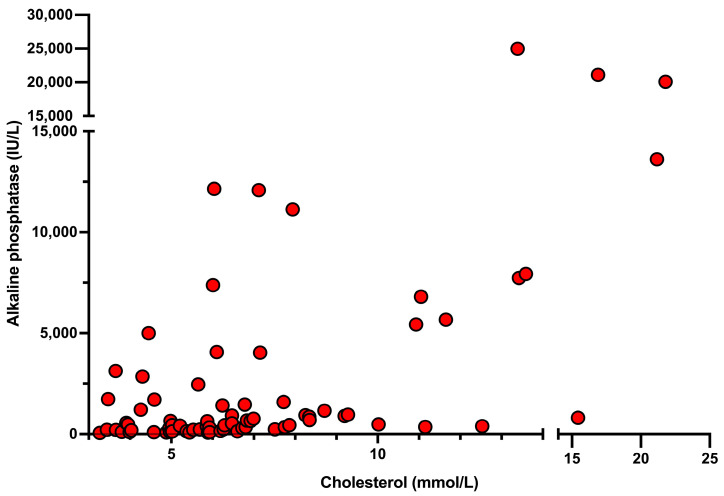
Correlation of alkaline phosphatase activity and cholesterol concentration in 74 dogs with suspect acute pancreatitis (r_s_ = 0.498, *p* < 0.001).

**Table 1 animals-14-03281-t001:** Age, sex distribution, and body weight in suspect acute pancreatitis dogs including dogs without hypercholesterolaemia (NC group) and dogs with hypercholesterolaemia (HC group). Results are presented as median (IQR).

	All Dogs	NC Group	HC Group
**Number of Dogs**	74	41 (55.4%)	33 (44.6%)
**Sex**	38 females 36 males	21 females 20 males	17 females 16 males
**Age (years)**	8.0 (6.0–11.0)	9.0 (6.0–11.0)	8.0 (6.0–11.8)
**Body Weight (kg) ***	13.0 (7.9–24.1)	16.9 (7.6–27.4)	11.4 (7.9–23.1)
**Body Condition Score (out of 5) ****	3 (3–4)	3 (2–3)	4 (3–5)

* Body weight was only available for 66 (35 NC and 31 HC) dogs. ** Body condition score was only available for 42 (21 NC and 21 HC) dogs.

**Table 2 animals-14-03281-t002:** Cholesterol and triglyceride concentrations, alkaline phosphatase (ALP) and gamma-glutamyl transferase (GGT) activities, total bilirubin and quantitative canine pancreatic lipase concentrations and 1,2-o-dilauryl-rac-glycero-3-glutaric acid-(6′-methylresorufin) ester (DGGR) lipase activity in suspect acute pancreatitis dogs with hypercholesterolaemia (HC group) or without hypercholesterolaemia (NC group). Results are presented as median (interquartile range).

	All Dogs (*n* = 74)	NC Group (*n* = 41)	HC Group (*n* = 33)	Reference Interval
**Cholesterol** **(mmol/L)**	6.26 (4.98–8.27)	5.02 (4.15–5.93)	8.35 (7.14–12.09)	3.2–6.5
**Triglycerides** **(mmol/L)**	0.85 (0.62–1.55)	0.64 (0.56–0.99)	1.36 (0.88–2.00)	0.11–1.69
**ALP Activity** **(U/L)**	640 (231–2557)	380 (135–1312)	932 (461–7271)	0–82
**GGT Activity** **(U/L) ***	5 (1–12)	4 (1–8)	5 (1–29)	0–16
**Total Bilirubin** **(**μ**mol/L)**	5.1 (4.1–11.3)	5.0 (4.2–7.9)	5.1 (3.1–32.2)	0–10
**Quantitative cPL (**μ**g/L) ****	641 (438–1037)	709 (524–1074)	546 (412–1108)	<200
**DGGR Lipase Activity (IU/L)**	121.6 (55.4–209.3)	132.0 (62.5–231.4)	101.0 (48.7–183.5)	<130

* GGT activity was only measured in 73 (40 NC and 33 HC) dogs. ** Quantitative cPL was only measured in 62 (33 NC and 29 HC) dogs.

## Data Availability

The datasets analysed during the current study are available from the corresponding author on reasonable request.

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
