# Peer review of "Association Between Hyperlipidaemia and Selected Cholestatic Markers in 74 Dogs with Suspect Acute Pancreatitis"

_animals, 2024, doi:10.3390/ani14223281_

Round 1

Reviewer 1 Report

Comments and Suggestions for Authors

The authors carried out an interesting retrospective study, but with some points that need to be better described for proper understanding and robustness of the data. The conclusion would need to be rewritten to match what was presented.

Simple summary

- include “serum”, “plasma” or blood to make it clear that the tests are blood tests

- This phrase “the higher the ALP and GGT 18 activities, the higher the cholesterol concentration” is redundant with “There was a significant positive correlation between the cholesterol concentration and ALP and gamma-glutamyl transferase (GGT) activities”. By improving the phrase, you gain space to include more information.

- I would change "provide evidence" to "strengthen the hypothesis" or something similar, because I don't think it's evidence in itself, but for now it's still preliminary data.

At many points "Abstract" and "Simple summary" are repetitive. To make them less repetitive, could you include more information to replace lines 33-34 "other cholestatic markers" with more data, for example what these "other cholestatic markers" are?  

Material and methods

- line 62 – Were included only positive qualitative cPL? Make it clearer because not all the animals had quantitative cPL measured, right?

- line 70-71 – it was not clear whether dogs with hypocholesterolemia per se or hypocholesterolemia due to medication use were excluded. In my opinion, it makes sense to exclude those on medication, but hypocholesterolemics per se don't make sense

- Line 70 – Regarding pre-existing diseases, was it considered in this case whether all the animals had been investigated for endocrine diseases such as hypercortisolism, diabetes mellitus and/or hypothyroidism? I understand that, as this is a retrospective study, this can be complicated, but it is important to go into more detail on this issue of pre-existing diseases in the material and methods, in the results (line 85) and to discuss the fact that endocrinopathies are predisposing factors for hypercholesterolemia, pancreatitis and also cholestasis. Everything can be interconnected, but not necessarily. So it's a bias in the study that needs to be addressed.

Line 76 – IQR – write in full, In this line and in the title of table 2

Results:

Line 94 – weight itself doesn't provide any information. The body condition score should be addressed. In the absence of this information, it should be discussed

Line 102 – So, all HC dogs had increased ALP, right? Perhaps writing in this way will make the information clearer.

Line 103-104 – confuse. What was different from what? O ideal seria comparar ALP activity in HC x NC.

Line 104 – You've used the acronym before, so you don't need to write it out in full at this point in the text.

Line 105-107 – in my opinion, at the least in correlation analyses, the data of hypocholesterolemics (without medication) should be included

Figure 1 – incluiria na legenda do gráfico o resultado de r e p.

Why is there only a figure for the ALP? You could have the figure for the other biomarkers of cholestasis.

Line 117-118 – Authors could have described all the variables for these four animals in a simple way in the same parentheses.

Lina 119-126 – Couldn't these two variables be included in the table? It would standardize the presentation of the data.

Discussion

- Line 143-144 – Couldn't this discrepancy be due to the exclusion of hypocholesterolemics? Please include the hypocholesterolemic animals in the study or discuss very carefully why they were excluded.

Line 147 – “strict exclusion criteria” – detail in the material and methods what these strict criteria were because this was not detailed. Should all the animals have had a suppression test, for example?

Line 152-154 – include references

Line 156-157 – include references

Line 163 – as this prevalence has not been presented before, I suggest here putting the exact result 22.7% (17/75) and replacing "prevalence" with "frequency" because no study has been carried out to determine the epidemiological context of prevalence.

Line 165 – explain why you're using the 5 mmol/L cutoff, including references.

Line 170 – better discuss what DGGR is and its importance, as well as cPL, justifying or creating hypotheses for their results.

Conclusion

“suggest that cholestasis is a likely cause of the hypercholesterolaemia in acute pancreatitis” – I'm not sure you can say that. It's not possible to create a relationship between cause and effect from the design. You can only say that there is correlation between the data or significance between them and not cause and effect.

Line 187 - generally mild – avoid generic terms like these, especially in the conclusion. "Generally" and "mild" are generic because they don't provide any scientific information. The study did not define what "mild" hypertriglyceridemia is.

Author Response

We would like to thank Reviewer 1 for their inciteful input and we feel that the changes we have made significantly increase the clarity of our study.

Simple summary

Comment 1: include “serum”, “plasma” or blood to make it clear that the tests are blood tests

Response 1:The word circulating has been included and other cholestatic markers listed as follows

Lines 12-14:  The aim of this study was to investigate the association between hyperlipidaemia and other circulating markers of cholestasis (alkaline phosphatase (ALP) and gamma-glutamyl-transferase (GGT) activities) in dogs with acute pancreatitis

Comment 2: This phrase “the higher the ALP and GGT 18 activities, the higher the cholesterol concentration” is redundant with “There was a significant positive correlation between the cholesterol concentration and ALP and gamma-glutamyl transferase (GGT) activities”. By improving the phrase, you gain space to include more information.

Response 2: The sentence “the higher the ALP and GGT activities, the higher the cholesterol concentration” has been removed.

Comment 3: I would change "provide evidence" to "strengthen the hypothesis" or something similar, because I don't think it's evidence in itself, but for now it's still preliminary data.

Response 3: This has been changed as follows;

Lines 19-20:  Such results support the hypothesis that cholestasis plays a role in the development of hypercholesterolaemia in dogs with acute pancreatitis.

Comment 4: At many points "Abstract" and "Simple summary" are repetitive. To make them less repetitive, could you include more information to replace lines 33-34 "other cholestatic markers" with more data, for example what these "other cholestatic markers" are?  

Response 4: This has been modified as follows

Lines 33-34: Cholesterol concentration was correlated with ALP and GGT activities suggesting an association between cholestasis and hypercholesterolaemia in dogs with acute pancreatitis.

Material and methods

Comment 5: line 62 – Were included only positive qualitative cPL? Make it clearer because not all the animals had quantitative cPL measured, right?

Response 5: We realize this may be unclear and have changed this to clarify that a diagnosis of acute pancreatitis was based on ultrasonographic evidence or a cPL > 400 together with clinician diagnosis and acute treatment. In reviewing this we realized, one case did not fulfill all the requirements and this has subsequently been removed and the data changes accordingly. We have also added to the Materials & Methods, how the cPL measurement is dealt with.

See below;

Lines 63-68: In these cases, results are classified as normal (corresponding to a quantitative cPL concentration of < 200 μg/L), abnormal (quantitative cPL ≥ 200 μg/L) or equivocal (not possible to confidently place into either of these categories).  With either of the latter two results, quantitative cPL measurement is carried out (Spec cPL™, IDEXX Laboratories Inc.).  If the point-of-care CPL was not recorded, the case was only included if a quantitative cPL result was available.

Lines 69-74: A diagnosis of suspect acute pancreatitis was made if there were either ultrasonographic changes suggesting acute pancreatitis (i.e. pancreatic enlargement, hypoechogenicity of the parenchyma, ill-defined margins, hypoechoic stripes and anechoic lesions, hyperechogenicity of the peripancreatic fat) or, the quantitative cPL was 400µg/L and both the clinician-stated diagnosis was pancreatitis and treatment qualified the disease as acute (intravenous fluid therapy, analgesia), as previously described (Hope et al. 2021).

Comment 6: line 70-71 – it was not clear whether dogs with hypocholesterolemia per se or hypocholesterolemia due to medication use were excluded. In my opinion, it makes sense to exclude those on medication, but hypocholesterolemics per se don't make sense

Response 6: We excluded dogs with hypocholesterolaemia because of concurrent disorders associated with hypocholesterolaemia, in much the same way as we dealt with disorders/drugs causing hyperlipidaemia.  We have clarified this as follows;

Lines 75-80: Dogs with incomplete biochemical data were excluded. Dogs that had pre-existing disorders or drug therapies associated with hyperlipidaemia or hypocholesterolaemia were also excluded because of the possibility of exacerbating or masking changes associated with pancreatitis.

Comment 7: Line 70 – Regarding pre-existing diseases, was it considered in this case whether all the animals had been investigated for endocrine diseases such as hypercortisolism, diabetes mellitus and/or hypothyroidism? I understand that, as this is a retrospective study, this can be complicated, but it is important to go into more detail on this issue of pre-existing diseases in the material and methods, in the results (line 85) and to discuss the fact that endocrinopathies are predisposing factors for hypercholesterolemia, pancreatitis and also cholestasis. Everything can be interconnected, but not necessarily. So it's a bias in the study that needs to be addressed.

Response 7: We agree that this is a potential bias in the study.  However, apart from diabetes mellitus, which is easy to recognize, other endocrine disorders did not specifically have to be ruled out and indeed such investigations have their own challenges in acute non-endocrine illness.  We have added a sentence to the Materials and Methods to explain this and have also expanded the discussion on limitations as follows;

Lines 78-80:  Specifically testing all dogs for non-pancreatitis causes of lipid abnormalities was not required but was at the discretion of the attending clinician.

Lines 199-206: There are several limitations of this study not least its retrospective nature. Further diagnostic investigations to rule in or out concurrent disorders were at the discretion of the attending clinician. Thus, a concurrent cause of hyperlipidaemia could not be completely excluded for all cases. In particular, dogs were not always specifically investigated for endocrine disorders such as hypothyroidism or hypercortisolism that can be associated with hypertriglyceridaemia, cholestasis and pancreatitis. However there appeared to be no clinical indication for such investigations that may in themselves be influenced by the acute nature of such an illness as pancreatitis.

Comment 8: Line 76 – IQR – write in full, In this line and in the title of table 2

Response 8: This was modified as indicated

Lines 83-85: Data were reported as median (interquartile range, IQR) or mean ± standard deviation (sd), as appropriate.

Results:

Comment 9: Line 94 – weight itself doesn't provide any information. The body condition score should be addressed. In the absence of this information, it should be discussed

Response 9: Body condition score was added to the results and relevant table.  As there was a significant difference this has also been discussed;

Lines 105-106: Body condition score was significantly (p < 0.0001) higher in the HC compared to the NC group.

Lines 164-168: Body condition score was significantly higher in hypercholesterolaemic dogs.  However, although obesity is associated with hypercholesterolaemia, the magnitude of change and its association with other cholestatic abnormalities makes obesity an unlikely explanation for the changes observed in the current study. In addition, body condition score was not available for all dogs in each of the two groups.

Comment 10: Line 102 – So, all HC dogs had increased ALP, right? Perhaps writing in this way will make the information clearer.

Response 10: This has been modified for clarification

Lines 113-116: All dogs in the HC group had increased ALP activity. None of the remaining four dogs with an ALP activity within reference interval had hypercholesterolaemia. The proportion of dogs with increased ALP activity was not significantly different between the two groups (p = 0.065).

Comment 11: Line 103-104 – confuse. What was different from what? O ideal seria comparar ALP activity in HC x NC.

Response 11: This has been modified as described in number 10 above.

Comment 12: Line 104 – You've used the acronym before, so you don't need to write it out in full at this point in the text.

Response 12: This has been modified as requested.

Comment 13: Line 105-107 – in my opinion, at the least in correlation analyses, the data of hypocholesterolemics (without medication) should be included.

Response 13: Please see the explanation in number 6 above.

Comment 14: Figure 1 – incluiria na legenda do gráfico o resultado de r e p.

Response 14: We think this means that you would like significance and r/p values in figure legend and this has been added.

Comment: Why is there only a figure for the ALP? You could have the figure for the other biomarkers of cholestasis.

Response 15: We think that they are superfluous to the information in the text as they add little in terms of correlation. If the Editor feels they are worthwhile, we can add them in.

Comment 16: Line 117-118 – Authors could have described all the variables for these four animals in a simple way in the same parentheses.

Response 16: This has been modified as follows

Lines 129-132: Four dogs had extrahepatic bile duct obstruction and all had severe hypercholesterolaemia (range: 13.59 – 21.79 mmol/L), marked hyperbilirubinaemia (range: 81.1 – 183.8 mmol/L), and severe increases in ALP (range: 7935 - 21,120 IU/L) and GGT (range: 41 – 189 IU/L) activities.

Comment 17: Lina 119-126 – Couldn't these two variables be included in the table? It would standardize the presentation of the data.

Response 17: These have been added to the table and the text modified accordingly.

Discussion

Comment 18: Line 143-144 – Couldn't this discrepancy be due to the exclusion of hypocholesterolemics? Please include the hypocholesterolemic animals in the study or discuss very carefully why they were excluded.

Response 18: Please see the explanation in number 6 above. Also we have added that this may have influenced our results in the discussion.

Lines 161-164: In addition, in the present study, excluding dogs with concurrent disorders associated with hypocholesterolaemia may also influence results as such exclusion criteria are not always applied in other studies [1].

Comment 19: Line 147 – “strict exclusion criteria” – detail in the material and methods what these strict criteria were because this was not detailed. Should all the animals have had a suppression test, for example?

Response 19: Please see the explanation in 7 above.

Comment 20: Line 152-154 – include references

Response 20: Added reference number 11.

Comment 21: Line 156-157 – include references

Response 21: Added reference number 12.

Comment 22: Line 163 – as this prevalence has not been presented before, I suggest here putting the exact result 22.7% (17/75) and replacing "prevalence" with "frequency" because no study has been carried out to determine the epidemiological context of prevalence.

Response 22: We have added this percentage to the results. We have not changed prevalence as it is routine terminology for a study such as this as it indicates the frequency of the abnormality in the disease.  If the Editor wishes to change, we are happy to do so.

Lines 108-109: Triglyceride concentrations were increased in 17 (23.0 %) dogs.

Comment 23: Line 165 – explain why you're using the 5 mmol/L cutoff, including references.

Response 23: We have added reference 13 and clarified the use of this cu-off

Lines 185-188: However, as in the previous study, the increases were marginal, with no value exceeding 5 mmol/L reflecting the concentration often used to approximate the therapeutic target in hypertriglyceridaemia [13]. This further supports that the hypertriglyceridaemia associated with acute pancreatitis is mild.

Comment 24: Line 170 – better discuss what DGGR is and its importance, as well as cPL, justifying or creating hypotheses for their results.

Response 24: We have added a broad explanation.

Lines 192-194: Both cPL concentrations and DGGR lipase activity are used as markers of pancreatitis with similar diagnostic performance with neither having perfect sensitivity or specificity for acute disease [14].

Comment 25: Conclusion “suggest that cholestasis is a likely cause of the hypercholesterolaemia in acute pancreatitis” – I'm not sure you can say that. It's not possible to create a relationship between cause and effect from the design. You can only say that there is correlation between the data or significance between them and not cause and effect.

Response 25: The conclusion has been modified as follow:

Lines 216-217: In conclusion, the results of this study suggest that there is an association between cholesterol concentration and markers of cholestasis in dogs with acute pancreatitis.

Comment 26: Line 187 - generally mild – avoid generic terms like these, especially in the conclusion. "Generally" and "mild" are generic because they don't provide any scientific information. The study did not define what "mild" hypertriglyceridemia is.

Response 26: “Generally” was removed and mild now has an explanation outlined above in number 23.

Reviewer 2 Report

Comments and Suggestions for Authors

1. What is the direct mechanistic link between cholestasis and hypercholesterolaemia in dogs with acute pancreatitis?

The study shows a moderate correlation between ALP activity and cholesterol levels but does not explain how cholestasis might directly cause hypercholesterolaemia in dogs with acute pancreatitis. The data suggest an association, but no mechanistic insight is provided to explain the observed correlations.

The authors should perform additional analyses or design further experiments to explore how cholestasis may mechanistically lead to hypercholesterolaemia in these dogs. Exploring the role of bile acids, liver function, or lipid metabolism pathways could provide more depth to these findings.

2. Why is the correlation between cholesterol and GGT weaker compared to ALP?

The results indicate a weaker correlation between cholesterol and GGT than between cholesterol and ALP. The manuscript does not fully explain why this difference exists, especially given that both ALP and GGT are markers of cholestasis.

The authors should clarify the physiological reasons behind this weaker correlation or conduct further analysis to explore whether the differential sensitivity of these markers in relation to hypercholesterolaemia is due to the specific stages of cholestasis or the role each marker plays in the liver and biliary system. Further exploration into the pathophysiology of GGT versus ALP in cholestasis may be warranted.

3. Why is there no significant correlation between cholesterol levels and quantitative cPL or DGGR lipase activity?

The paper reports that there is no correlation between cholesterol levels and quantitative canine pancreatic lipase (cPL) or DGGR lipase activity, which raises questions about the relationship between pancreatic inflammation and lipid abnormalities. The paper does not offer a clear explanation for this lack of association.

The authors should investigate the pathophysiological basis for why pancreatic lipase levels (cPL and DGGR) do not correlate with cholesterol concentrations in this cohort of dogs. Further studies could explore whether the degree of pancreatitis severity or chronicity, rather than acute pancreatitis alone, impacts lipid metabolism differently.

4. How do different severities of pancreatitis affect lipid metabolism in these dogs?

The study does not differentiate between different severities of acute pancreatitis, and it is unclear whether the degree of pancreatitis influences hyperlipidaemia or cholestasis markers differently. Grouping all cases of suspected pancreatitis together may obscure important relationships.

The authors should stratify the pancreatitis cases by severity and reanalyze the data to determine whether more severe cases of pancreatitis show stronger associations with hypercholesterolaemia or different patterns of cholestatic markers. This could provide more detailed insights into the clinical relevance of lipid abnormalities. 

Minor Concerns:

The inclusion of more detailed information on the dogs' clinical presentations, treatments, and outcomes could enhance the understanding of the study's context and findings.

Author Response

We would like to thank Reviewer 2 for their constructive criticisms.  We have attempted to modify the manuscript as much as possible but some changes are not possible because of the retrospective nature of this study.  We feel that we have outlined these in the limitations section.  

Reviewer 2

Comment 1: What is the direct mechanistic link between cholestasis and hypercholesterolaemia in dogs with acute pancreatitis?

The study shows a moderate correlation between ALP activity and cholesterol levels but does not explain how cholestasis might directly cause hypercholesterolaemia in dogs with acute pancreatitis. The data suggest an association, but no mechanistic insight is provided to explain the observed correlations.

The authors should perform additional analyses or design further experiments to explore how cholestasis may mechanistically lead to hypercholesterolaemia in these dogs. Exploring the role of bile acids, liver function, or lipid metabolism pathways could provide more depth to these findings.

Response 1: We thank the reviewer for their comments but much of this was not within the scope of this study.  We have amended the conclusion to emphasise that an association was found but that further research is required to investigate possible pathophysiological mechanisms.

Comment 2: Why is the correlation between cholesterol and GGT weaker compared to ALP?

The results indicate a weaker correlation between cholesterol and GGT than between cholesterol and ALP. The manuscript does not fully explain why this difference exists, especially given that both ALP and GGT are markers of cholestasis.

The authors should clarify the physiological reasons behind this weaker correlation or conduct further analysis to explore whether the differential sensitivity of these markers in relation to hypercholesterolaemia is due to the specific stages of cholestasis or the role each marker plays in the liver and biliary system. Further exploration into the pathophysiology of GGT versus ALP in cholestasis may be warranted.

Response 2: We have provided an explanation.

Lines 174-176: Gamma-glutamyl transferase is considered to be a less sensitive marker of hepatobiliary disease in dogs and increases are of a lesser magnitude compared with those in ALP activity [11].

Comment 3: Why is there no significant correlation between cholesterol levels and quantitative cPL or DGGR lipase activity?

The paper reports that there is no correlation between cholesterol levels and quantitative canine pancreatic lipase (cPL) or DGGR lipase activity, which raises questions about the relationship between pancreatic inflammation and lipid abnormalities. The paper does not offer a clear explanation for this lack of association.

The authors should investigate the pathophysiological basis for why pancreatic lipase levels (cPL and DGGR) do not correlate with cholesterol concentrations in this cohort of dogs. Further studies could explore whether the degree of pancreatitis severity or chronicity, rather than acute pancreatitis alone, impacts lipid metabolism differently.

Response 3: The scope of this paper was to explore a potential association and to carry out further in depth studies would require a much more detailed prospective study. We have added further explanation about the fact

Lines 192-198. Both cPL concentrations and DGGR lipase activity are used as markers of pancreatitis with similar diagnostic performance with neither having perfect sensitivity or specificity for acute disease [14]. There was no correlation between quantitative cPL concentration or DGGR lipase activity with cholesterol concentration. Quantitative cPL concentration and DGGR lipase activity have been shown to be poor indicators of severity, clinical progression or survival in dogs with acute pancreatitis [14, 15], possibly explaining any lack of association.

Comment 4: How do different severities of pancreatitis affect lipid metabolism in these dogs?

The study does not differentiate between different severities of acute pancreatitis, and it is unclear whether the degree of pancreatitis influences hyperlipidaemia or cholestasis markers differently. Grouping all cases of suspected pancreatitis together may obscure important relationships.

The authors should stratify the pancreatitis cases by severity and reanalyze the data to determine whether more severe cases of pancreatitis show stronger associations with hypercholesterolaemia or different patterns of cholestatic markers. This could provide more detailed insights into the clinical relevance of lipid abnormalities. 

Response 4: We recognize that assessment of severity of pancreatitis is a limitation and have added this into the limitations and an area for future study

Lines 206-207: No attempt was made to assess severity of disease because of the complexities associated with its retrospective nature.

Lines 217-219: Further studies are warranted to investigate the impact of severity and the underlying pathophysiological mechanisms explaining this association.

Comment 5: Minor Concerns: The inclusion of more detailed information on the dogs' clinical presentations, treatments, and outcomes could enhance the understanding of the study's context and findings.

Response 5: This was a retrospective study designed to investigate an association between cholesterol and other cholestatic markers.  It would be beyond the scope of the study to provide any more information.

Round 2

Reviewer 1 Report

Comments and Suggestions for Authors

The authors modified the article based on the suggestions made. They left two minor points to be decided by the editor. As a way of further improving the article, I leave some small suggestions below.

- line 65 – It's still a bit confusing. “equivocal (not possible to confidently place into either of these categories)”. If it's a quantitative test, how could it not be classified? Just explain that in text.

- line 67 – cPL (lower case letter)

- table 2 – include * and make it clear how many animals took each measurement, especially when there weren't 74 (in the case of GGT), as was done in table 1.

Author Response

Reviewer 1:

The authors modified the article based on the suggestions made. They left two minor points to be decided by the editor. As a way of further improving the article, I leave some small suggestions below.

Comment 1: line 65 – It's still a bit confusing. “equivocal (not possible to confidently place into either of these categories)”. If it's a quantitative test, how could it not be classified? Just explain that in text.

Response 1: Amended

Lines 61-63: Case records of dogs seen between January 2014 and June 2022 that had a qualitative point-of-care canine pancreatic lipase (cPL) (SNAP cPL™, IDEXX Laboratories) charged were retrospectively reviewed.”

Comment 2: line 67 – cPL (lower case letter)

Response 2: Corrected.

Comment 3: table 2 – include * and make it clear how many animals took each measurement, especially when there weren't 74 (in the case of GGT), as was done in table 1.

Response 3: Added to table 2 for GGT activity and quantitative cPL as follow: lines 149-151

“*GGT activity was only measured in 73 (40 NC and 33 HC) dogs.

***Quantitative cPL was only measured in 62 (33 NC and 29 HC) dogs.”

Reviewer 2 Report

Comments and Suggestions for Authors

The authors have addressed my concerns

Author Response

Reviewer 2:

The authors have addressed my concerns.

Response: The authors would like to thank the reviewer.